# Decoding Immuno-Competence: A Novel Analysis of Complete Blood Cell Count Data in COVID-19 Outcomes

**DOI:** 10.3390/biomedicines12040871

**Published:** 2024-04-15

**Authors:** Prakasha Kempaiah, Claudia R. Libertin, Rohit A. Chitale, Islam Naeyma, Vasili Pleqi, Johnathan M. Sheele, Michelle J. Iandiorio, Almira L. Hoogesteijn, Thomas R. Caulfield, Ariel L. Rivas

**Affiliations:** 1Department of Medicine, Division of Infectious Diseases, Mayo Clinic, Jacksonville, FL 32224, USA; prakashamk@gmail.com (P.K.); pleqi.vasili@mayo.edu (V.P.); 2Department of Critical Care, Mayo Clinic, Jacksonville, FL 32224, USA; idxpert@gmail.com; 3Mayo Clinic Alix School of Medicine, Mayo Clinic, Jacksonville, FL 32224, USA; chitale.rohit@mayo.edu; 4Department of Neuroscience, Division of QHS Computational Biology, Mayo Clinic, Jacksonville, FL 32224, USA; islam.naeyma@mayo.edu (I.N.); caulfield.thomas@mayo.edu (T.R.C.); 5Department of Emergency Medicine, Mayo Clinic, Jacksonville, FL 32224, USA; sheele.johnathan@mayo.edu; 6Department of Internal Medicine, School of Medicine, University of New Mexico, Albuquerque, NM 87131, USA; miandiorio@salud.unm.edu; 7Human Ecology, Centro de Investigaciones Avanzadas, Merida 97310, Mexico; almirahoo@cinvestav.mx; 8Department of Biochemistry and Molecular Biology, Mayo Clinic, Rochester, MN 55905, USA; 9Center for Global Health, Department of Internal Medicine, School of Medicine, University of New Mexico, Albuquerque, NM 87131, USA

**Keywords:** immune-competence, infectious disease, COVID-19

## Abstract

Background: While ‘immuno-competence’ is a well-known term, it lacks an operational definition. To address this omission, this study explored whether the temporal and structured data of the complete blood cell count (CBC) can rapidly estimate immuno-competence. To this end, one or more ratios that included data on all monocytes, lymphocytes and neutrophils were investigated. Materials and methods: Longitudinal CBC data collected from 101 COVID-19 patients (291 observations) were analyzed. Dynamics were estimated with several approaches, which included non-structured (the classic CBC format) and structured data. Structured data were assessed as complex ratios that capture multicellular interactions among leukocytes. In comparing survivors with non-survivors, the hypothesis that immuno-competence may exhibit feedback-like (oscillatory or cyclic) responses was tested. Results: While non-structured data did not distinguish survivors from non-survivors, structured data revealed immunological and statistical differences between outcomes: while survivors exhibited oscillatory data patterns, non-survivors did not. In survivors, many variables (including IL-6, hemoglobin and several complex indicators) showed values above or below the levels observed on day 1 of the hospitalization period, displaying L-shaped data distributions (positive kurtosis). In contrast, non-survivors did not exhibit kurtosis. Three immunologically defined data subsets included only survivors. Because information was based on visual patterns generated in real time, this method can, potentially, provide information rapidly. Discussion: The hypothesis that immuno-competence expresses feedback-like loops when immunological data are structured was not rejected. This function seemed to be impaired in immuno-suppressed individuals. While this method rapidly informs, it is only a guide that, to be confirmed, requires additional tests. Despite this limitation, the fact that three protective (survival-associated) immunological data subsets were observed since day 1 supports many clinical decisions, including the early and personalized prognosis and identification of targets that immunomodulatory therapies could pursue. Because it extracts more information from the same data, structured data may replace the century-old format of the CBC.

## 1. Introduction

To prevent the spread of rapidly disseminating infections, such as COVID-19, personalized assessments of immunity have been proposed [1]. Yet, immuno-competence lacks a standardized method that could routinely be used in clinical medicine [2]. Tests that measure immuno-competence should be both informative and biologically valid. Ideally, they should also predict disease outcomes and help select personalized therapies [3,4]. To develop such methods, several biomedical and methodological concepts need to be considered and integrated into the chosen operationalizations, including (i) *dynamics*, (ii) *non-binary* (polychotomous) conditions, (iii) the risk of *confounding*, (iv) *personalized* factors, (v) biological *complexity*, (vi) *data structuring* and (vii) *data analysis*.

### 1.1. Dynamic, Polychotomous and Personalized Methods That Prevent Confounding

The dynamics of the immune responses, such as COVID-19, are poorly known [5]. One reason for such a cognitive gap is that, typically, classic methods have been static. Static methods assume only two conditions: “disease-positive” and “disease-negative” results. Binary and static methods are not valid because biological functions are dynamic processes that can reveal three or more biomedical conditions [6,7]. For example, inflammatory (immunological) responses exhibit three or more presentations: no inflammation, early inflammation, late inflammation (followed by recovery) and/or late inflammation (followed by chronic disease). Therefore, to assess immuno-competence, new methods should handle non-dichotomous and dynamic conditions [8,9,10]. By subtracting earlier from later data values, the potential bias of extremely low (or high) values of earlier tests can be prevented, and dynamics can be assessed [11].

In addition, assessments of immuno-competence should be personalized. Because patients differ in co-morbidities and medical histories, to prevent *confounding*, methods that attempt to capture dynamic and non-binary responses should not assume that populations are homogenous [12].

Because most published research on major diseases, such as COVID-19, has been cross-sectional and observational, the risk of bias is not trivial [13,14,15,16]. One source of confounding refers to the fact that patients may be hospitalized at different disease stages, and therefore, early and late disease stages could erroneously be grouped together. Another variation of the same error can occur when data with different disease stages are aggregated. Aggregate data do not provide personalized information but population averages. Thus, methods that handle non-binary, dynamic and personalized data are needed [17,18].

A third common source of confounding emerges when data distributions of different biological conditions—such as “cases” and “non-cases”—overlap [19,20]. *Data overlapping* (different outcomes associated with similar data intervals) should be avoided because it prevents medical discrimination even when statistical significance is achieved [21,22]. The previous considerations result in one consequence that novel methods should also address: given the dynamic, non-binary and personalized aspects of disease processes, there is no standard patient, and therefore, there is no control either [23].

### 1.2. Complex and Dynamic Interactions That Structure the Data and Show Patterns

To measure personalized immune dynamics and to extract hidden patterns, the assessment of immunological complexity has been recommended [24,25,26]. To explore complexity, interactions involving two or more biological elements—such as lymphocytes and monocytes—may describe biological functions that influence outcomes [27]. Such complex functions change over time and may exhibit feedforward and feedback loops [9,28].

*Structured* data—not isolated variables, such as the counts or percentages of a single cell type—can investigate relationships. Data structuring may provide interpretable and medically useful information and, in addition, avoid the confounding associated with unstructured data [29].

Structured data may promote pattern recognition. The *shapes of data* distributions may inform [30,31]. One specific data distribution (positive kurtosis or L-shaped data) has been viewed, for many years, as statistically intractable [32]. Yet, such data presentations are not rare and may be informative in several medical fields [33,34,35].

Interpretations also depend on the *format* used to present and analyze the data. Because complex and dynamic processes involve interactions, ratios—which assess relationships among two or more elements—are preferred over counts. Novel methods that utilize the data generated by the century-old CBC could inform more or better if they are not limited to analyses of counts [29,36].

### 1.3. COVID-19-Related Dynamic and Personalized Immuno-Competence

The previous considerations may influence the way immuno-competence is explored. In several clinical entities (e.g., COVID-19, sepsis, hantavirus), the early disease stage is not always associated with death [27,37,38,39,40].

Immuno-competence cannot be evaluated with a single (static) assessment. This concept may matter particularly in sepsis where dynamic changes can occur and both excessive inflammation and immuno-suppression may induce fatalities [41,42]. Therefore, to estimate immuno-competence, temporal assessments are needed.

In adopting a personalized approach, here, longitudinal data on COVID-19 are retrospectively explored with the purpose of assessing immunological dynamics and their associated outcomes. Assuming that survival is an unambiguous outcome of immuno-competent responses, this study was set to answer whether, soon after admission, (i) structured and temporal blood-related data can extract more information from the same data than unstructured data and (ii) whether non-overlapping and/or distinct data distributions may, rapidly and easily, predict disease outcomes in COVID-19 patients.

## 2. Materials and Methods

### 2.1. Clinical Data

Two-hundred and ninety-one complete blood count (CBC/differential) samples collected from 101 anonymized individuals hospitalized at the Mayo Clinic of Jacksonville, Florida before August of 2020 with a COVID-19 diagnosis are reported in Appendix A. Inclusion criteria considered subjects of 18 or more years of age, treated at or before August 2020, with SARS-CoV-2 positive test results conducted within 72 h of admission, radiographic changes consistent with COVID-19 and deemed to be at risk of severe illness. Discharge was based on negative testing and the alleviation of life-threatening conditions. Individuals treated for immunosuppression, malignancy, pregnancy and/or hospitalized for at least three weeks in the previous six months were excluded [27]. The outcome (survival or no survival) was determined at 30 days after hospitalization.

### 2.2. Data Structure and Analysis

Time was measured with seven approaches that analyzed uni-, bi- and three-dimensional data. The first approach analyzed the data collected on the first post-hospitalization day (Day 1). The second approach considered the data collected at days 1 and 2. The third approach evaluated the difference between day 2 and day 1 observations. These analyses were uni-dimensional and focused on a single (unstructured or structured) variable.

The fourth and fifth approaches were bi-dimensional, considering two (unstructured or structured) variables. The sixth and seventh approaches were three-dimensional, focusing on personalized and therapy-oriented information, respectively.

In all approaches, the presence or absence of L-shaped data distributions (positive kurtosis) was determined. Kurtosis (non-normally distributed [not bell-shaped] data) was documented when observations differed 5 or more standard deviations from the mean, and consequently, most observations were concentrated on one end of the distribution (hence, the shape of the data resembled an ‘L’). The statistical analysis was limited to indicating whether outcomes revealed non-kurtotic data intervals, and, when kurtosis was observed, whether L-shaped data distributions were revealed by one or many blood-related variables.

In addition, the informative effect of non-structured and structured data was also investigated. Observations were analyzed as either separate counts or relative percentages of single-cell types (unstructured approach) or complex multicellular interactions that included two or more ratios (structured approach). The structured method has been described before when several diseases that affect human and non-human species were investigated [21,24,27,28,40]. The structured method is based on dimensionless indicators (DIs)—temporary guides used to detect distinct data subsets. DIs include numerous relationships, and they are complex: they have two or more interactions that, individually, could include two or more elements, e.g., the triple ratio resulting from (i) generating a ratio between neutrophils and monocytes (the N% divided the M% or N/M ratio), (ii) creating a separate ratio between mononuclear cells (L%+M% or MC cells) and neutrophils (or MC/N ratio), and (iii) the overall relationship resulting from the first divided the second ratio. Because DLs are used to display three-dimensional patterns, the number of possible combinations observed is very high (if not infinite) at least because each axis may influence and/or be influenced by any other axis. Given such internal and external possible variability, DLs are identified with letters lacking any biological meaning (such as *AA*, *AAA*, *BBB*…). Biologically interpretable contents are explicitly investigated only after data patterns are detected in 3D space.

The rationale of this method can be found in Niels Jerne’s work [43]. He emphasized the *language*-like properties of immunology, i.e., a set of few or very few elements (comparable to the 26 letters of the English alphabet) which, once combined and recombined, results not only in a very large number of ‘words’, but also in an even larger number of ‘sentences’ and ‘paragraphs’. When we refer to Biology, this combinatorial ‘language’ can generate millions of antibodies and/or proteins that can recognize antigens even when the immune system has not previously encountered such antigens. 

This means that analyzing only the elemental immunological units is not informative: in considering the language metaphor, no knowledge can be derived from memorizing all the letters of the alphabet. Similarly, useful biomedical knowledge may be missed, unless and until basic leukocyte data are (a) reformatted as information, (b) information is converted into knowledge, and (c) finally, knowledge is applied as concrete (context- and resource-dependent) decisions. This process has been described as the ‘DIKW (data–information–knowledge–‘wisdom’) pyramid’ [29].

While a proprietary software can, in principle, conduct all these steps in real time, any method that focuses on pattern recognition and explores three-dimensional data combinations (structures in which the information emerges only after the three variables converge in space) is likely to detect usable information, as previously described [24].

This method was conducted with a proprietary software package [29,39,44]. To generate visual patterns and to assess statistical properties, a commercial software package was used (*Minitab 21*, Minitab LLC, State College, PA, USA).

## 3. Results

Due to overlapping data intervals of different outcomes, no medical discrimination was possible, at any time when either counts or percentages of cell types were assessed in isolation (rectangles, Figure 1A–D). Overlapping data distributions occurred even when statistically significant differences were observed. For example, when all the temporal data were considered, the median lymphocyte percentage of non-survivors was 12.3% (n = 37), while that of survivors was 17.2% (n = 254), a difference that reached statistical significance (*p* < 0.01, Mann–Whitney test, Appendix A). This statistical difference occurred even when most survivor and non-survivor observations overlapped (Figure 1D). 

The lack of discrimination between outcomes was not limited to leukocyte-related data. Other hematological variables—such as IL-6 and hemoglobin—also showed, over time, overlapping distributions between outcomes (Figure 2A,B). In contrast, structured data that simultaneously captured many interactions among all leukocytes displayed not only statistically significant differences between outcomes, but also a range of observations with non-overlapping outcomes (Figure 2C).

When, regardless of outcome, leukocytes were assessed as separate (non-structured) cell types at hospitalization day 1, no L-shaped (non-kurtotic) intervals were observed (Appendix A, Figure 3A,B). However, when other hematological variables—such as IL-6 and Hb—were assessed, survivors revealed kurtosis, while non-survivors did not (Appendix A, Figure 3C–F). Similarly, structured leukocyte data showed kurtosis, although only in survivors (*AAT*, Figure 3G,H).

Regardless of outcome, kurtosis was not observed when all longitudinal observations were assessed as non-structured data, i.e., when the percentages of lymphocytes, neutrophils or monocytes were investigated (Appendix A, Figure 4A–F). In contrast, both non-leukocyte and structured leukocyte data displayed kurtosis when survivors were tested and all temporal data points were evaluated (Figure 5A–F). Hence, it was concluded that kurtosis was associated with both (i) the *structure* of the data (complex ratios) and (ii) the *outcome* (survival).

To prevent biased assessments of dynamics, an additional comparison was conducted, which considered the net difference between two consecutive observations. Dynamics were then evaluated in two ways: (a) considering the net difference of values shown between day 2 and day 1 by either survivors or non-survivors and (b) considering the dispersion of such values when survivors and non-survivors were measured simultaneously. When day 1 values were subtracted from day 2 values, survivors displayed kurtosis, while non-survivors did not (Appendix A, Figure 6A–F). Some variables—such as IL-6 and complex indicators that assessed leukocyte multicellular interactions—oscillated over time in competent immunological responses. In contrast, the result of the subtraction, for most survivors, approached zero (Figure 6B,F). 

The last comparison simultaneously assessed both outcomes. Complementing the assessments of kurtosis reported in Figure 6, dynamic assessments supported the hypothesis that survivors displayed numerous and large temporal oscillations around zero, while non-survivors did not (Figure 7A–F).

To further explore the hypothesis that survivors and non-survivors differed immunologically, additional time-related, bi-dimensional comparisons between pairs of cell types were conducted with non-structured data. While survivors exhibited a broad range of observations that could facilitate relationships between neutrophils and monocytes at days 1 and 2, non-survivors lacked such data ranges (Figure 8A–D). Likewise, non-survivors did not show high values of lymphocyte percentages that could interact with monocytes at such times, while survivors did (Figure 8E–H).

More information was extracted from the same data when bi-dimensional, temporal and structured data were investigated. Two data ranges occupied only by survivors were revealed when complex leukocyte interactions were investigated at days 1 and 2 (Figure 9A–D). 

The two protective data ranges of immunological interactions were not influenced by time (Figure 10A–C). The two immunological relationships associated with survival (here named ‘A’ and ‘B’) were characterized by non-overlapping intervals of non-structured data (the percentages of lymphocytes [L], neutrophils [N] or monocytes [M]) and structured data (the small leukocytes [L and N]/M, the N/M and the N/L ratios, Figure 10D,E). 

When all temporal observations were considered (n = 291), three-dimensional assessments of structured data retrieved even more information from the data: three data subsets—perpendicular to one another—included only survivors when the first three hospitalization days were assessed (Figure 11A,B). Therefore, immuno-competence (responses that promoted survival) was an early process—it was measurable in the first three post-admission days. 

In classifying the data into four groups (three groups composed of only survivors [here named I, II and III] and one group that included survivors and non-survivors, Figure 11C), survivors revealed three non-overlapping data intervals. The data groups that included only survivors differed from one another: group I showed the lowest values of lymphocyte percentages and highest neutrophil/lymphocyte (N/L) ratios; group II showed the lowest monocyte values intermediate and N/L ratio values between those of groups I and III; and group III displayed the lowest neutrophil percentages (Figure 11D).

When day 1 and day 2 structured data were investigated (n = 195) as recorded (i.e., when the same patient contributed two data points), the findings supported at least four types of inferences. For example, between one and three data ranges concerning only a single outcome were detected at days 1 and 2 (Figure 12A,B). Because survival-related findings were based on two different data structures that considered two temporal points (a double redundant approach), inferences revealed internal validity (Figure 12A,B).

The three data subsets containing only survivors were named ‘left’, ‘top’ and ‘right’ (Figure 12C). They differed immunologically from one another: the same non-structured and structured variables that distinguished similar subsets when all temporal data were analyzed (reported in Figure 11) also differentiated the subsets identified with the data collected on the first two hospitalization days (Figure 12D). 

In addition, day 1 and day 2 structured data displayed trajectory: the directionality of the temporal data (where each data point came from/went to) was objectively displayed using an easily detectable structure (a single, one data point-wide, line of observations that simultaneously captures time and immunological interactions, Figure 12D). In this example, the arrows generated by this minimal (only 24 h long) temporal assessment identified two prognostic inferences: two movements away from zero predicted survival at a personalized level, i.e., even a single individual could be prognosticated.

Therefore, the findings reveal that (i) assessments limited to day 1 data are adequate for predicting outcomes; (ii) the data collected at day 1 and day 2 revealed trajectory; (iii) the assessment of dynamics (day 2 minus day 1 data) demonstrated that survivors’ data oscillated, while the data of non-survivors did not; and (iv) numerous hematological, non-cellular variables also revealed kurtosis-related differences between non-survivors and survivors. 

## 4. Discussion

### 4.1. Caveats

The fundamental limitation of this study refers to the fact that there is no consensus on the meaning of immuno-competence. To address this challenge, this study analyzed both humoral and cell-mediated responses. Because both approaches generated similar inferences (survivors exhibited L-shaped [kurtotic] data intervals while non-survivors did not), the stated limitation was ameliorated.

A second potential limitation is that, given the combinatorial nature of leukocyte-related multicellular interactions, the number of possible ratios to be assessed is very high, and consequently, any investigated ratio is not necessarily representative. The analysis of two or more leukocyte-derived complex indicators (redundant analysis) diminished this risk.

### 4.2. Seven Major Findings

The data supported seven inferences: (1) over time, survivors displayed non-randomly distributed (circular-like) immunological responses that revealed feedback/feedforward loops; (2) L-shaped data distributions (kurtosis) characterized survivors; (3) non-survivors were associated with static immunological responses; (4) immuno-competence (responses that, over time, differed in magnitude and/or internal composition) was expressed early, after hospitalization; (5) data ranges of immunological data associated with survival may provide targets for immunomodulatory strategies; (6) unstructured, simple (reductionist) indicators—such as counts and percentages—missed information conveyed by complex (non-reductionist) indicators; and (7) the anticipatory testing of immuno-competence (evaluations conducted in the absence of a disease) may be desirable. These inferences indicate that the time-related detection of non-randomly distributed data intervals may offer medically useful information.

These findings corroborate the non-random (oscillatory) data features of blood leukocytes. The circadian cycle of blood leukocytes has been described before—a process associated with immuno-competence [45]. For example, the circadian cycles of CD4 and CD8+ cells are opposed to one another [46]. The circadian cycle that regulates the hypothalamic–pituitary–thyroid axis is altered in immuno-suppression: in lung cancer, the ratio of melatonin/cortisol is diminished (cortisol is increased), and the count and relative proportions of CD8+ cells are decreased [47]. Such feedback-like cycles are not randomly distributed. For instance, lymphocyte data exhibit non-normally distributed intervals [48,49]. Because effective (immuno-competent) responses are non-random [50,51], a lack of such dynamics may reflect immuno-suppression or deficiency.

Here, we followed the new recommended research emphasis that promotes explorations of L-shaped data distributions, i.e., kurtosis [52,53]. In L-shaped distributions, data oscillations around zero (a quasi-stationary mode) induce a left-side peak [54]. Chronic infections have been characterized by a quasi-stationary mode in which lymphocytes are inactive [55]. Because non-normal (L-shaped or kurtotic) data distributions were mainly exhibited by survivors, the hypothesis that kurtosis describes cyclic immunological oscillations associated with survival was not rejected [9,24]. This continuous ‘fine-tuning’ resembles the way cars are driven: to drive straight, an open-ended series of minor turns to the left and right are needed. Other biological (non-immunological) functions also show similar cyclic adjustments [56].

Numerous studies have suggested that immuno-suppression is associated with COVID-19 [57,58,59,60,61,62,63]. This study also corroborated earlier reports that indicated immune-competence, in COVID-19, predicts survival [3]. The fact that, in patients with cancer, COVID-19 is associated with increased mortality supports the view that immuno-suppression is a negative predictor of disease outcome [64,65].

The findings also support the hypothesis that immuno-competence, in COVID-19, involves the cellular immune response [66,67,68,69]. Patients presenting with severe COVID-19 have frequently been reported with lymphopenia [70,71].

This study also showed differences between survivors and non-survivors when non-cellular factors (including hemoglobin, IL-6 and hematocrit) were analyzed [72]. Because numerous humoral indicators differed between the survivors and non-survivors and such differentiation was robust over time, the involvement of both arms of the immune system on disease outcomes was documented, and the approach used to explore immuno-competence exhibited robust internal and external validity [73].

Non-survivors showed differences in their immune responses as early as hospitalization day 1. Because the kurtosis associated with survivors was not detected after day 3, the findings support the hypothesis in that immuno-competence should be evaluated early [74]. Because immuno-suppression was already detected at hospitalization day 1, the hypothesis that, in non-survivors, immunosuppression could predate the infection cannot be ruled out [3]. This hypothesis has been sustained before: patients tested two or more weeks before SARS-CoV-2 infections took place have shown reduced numbers of eosinophils—a finding associated with poorer outcomes [75]. Very early immunological differences between survivors and non-survivors have also been reported in other studies [76,77,78]. 

Data ranges of immunological data associated with survival may provide targets for immunomodulatory therapies. Because, to be effective, immunomodulation should be implemented early, the findings provide unambiguous targets for such therapies [79].

In agreement with earlier reports, unstructured, simple indicators—such as counts and percentages—revealed weaknesses associated with reductionist approaches [21,80]. In contrast, structured (complex, non-reductionist) indicators provided biologically interpretable and medically useful information [24,26]. When complex data structures reveal distinct patterns—such as single, one data point-wide, lines of observations—the directionality of temporal data promotes earlier evaluations [81].

Because the risk of COVID-19-related death may be high even in vaccinated immuno-suppressed patients, the findings supported the implementation of the anticipatory testing of immuno-competence, i.e., before diseases are diagnosed [82]. This possibility may be synonymous with updating the format of the CBC/differential: after a century in which no biological functions (no interactions among cells and molecules that participate in immune responses) have been assessed, it is now possible to extract more biologically interpretable and medically useful information from the same data. 

## 5. Conclusions

This study suggests that the temporal assessment of non-randomly distributed, oscillatory cycles of immune responses may facilitate early and personalized prognosis as well as identification of targets to be potentially used by immunomodulatory-oriented therapies. To further explore this proposition, additional studies on dynamics of host–microbial interactions are recommended.

## Figures and Tables

**Figure 1 biomedicines-12-00871-f001:**
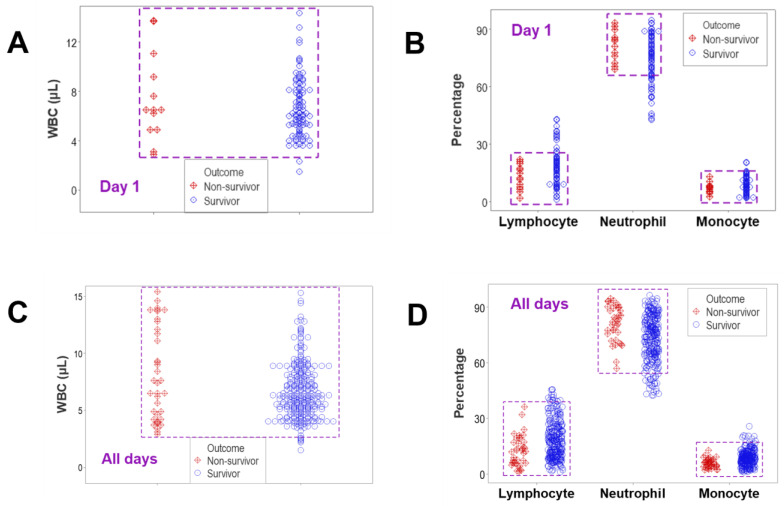
Non-structured data analysis. Total leukocyte counts (WBC) and relative percentages of lymphocytes, neutrophils and monocytes were assessed considering only the data available at hospitalization day 1 (**A**,**B**) and all temporal observations (**C**,**D**). In all comparisons, non-survivors and survivors displayed overlapping data distributions, which prevented their differentiation (rectangles, (**A**–**D**)).

**Figure 2 biomedicines-12-00871-f002:**
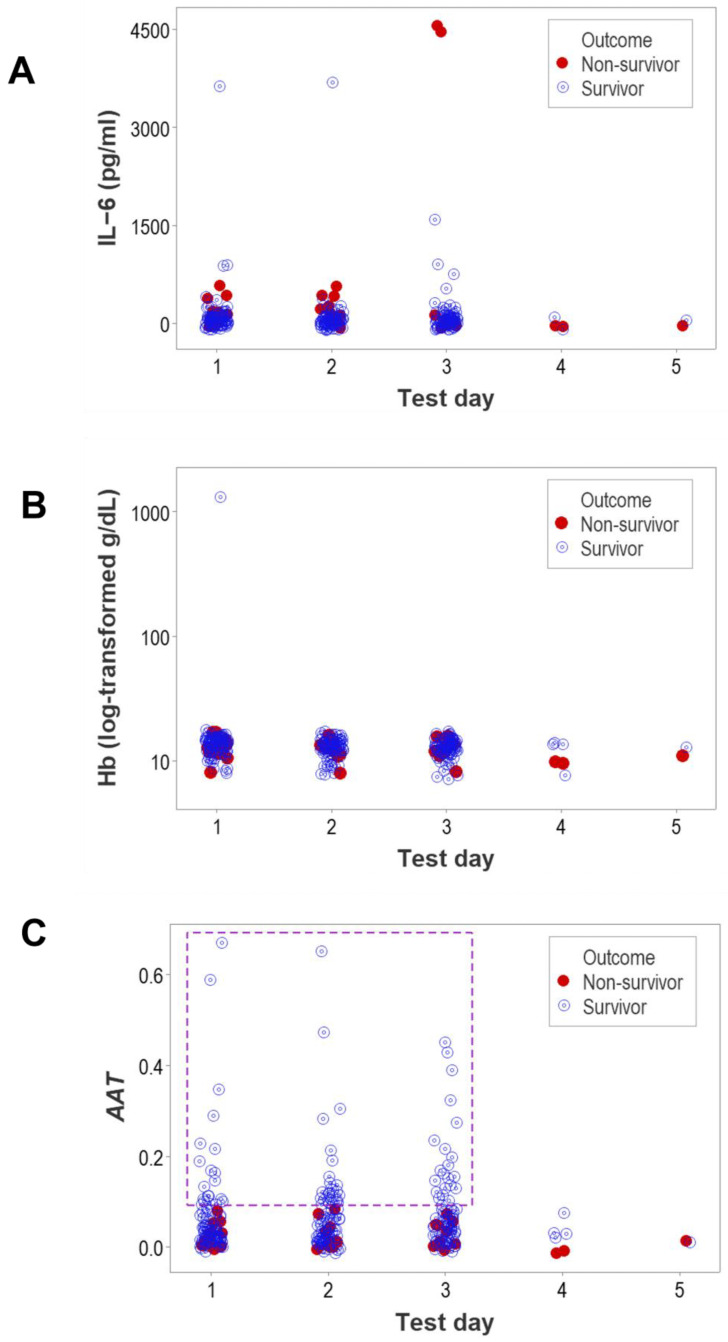
Day-specific, non-structured data analysis. The blood concentrations of IL-6 and hemoglobin (Hb) (**A**,**B**) as well as the values of a complex indicator that captured numerous relationships among leukocytes (**C**) were determined at each post-hospitalization day for non-survivors and survivors. The overlapping data distributions of IL-6 and Hb prevented the differentiation of outcomes (**A**,**B**). However, for the non-cellular, non-structured (IL-6, Hb) and leukocyte-related (AAT) structured variables in the first 3 days, survivors displayed a range of values for the complex indicator not including non-survivors (rectangle, (**C**)).

**Figure 3 biomedicines-12-00871-f003:**
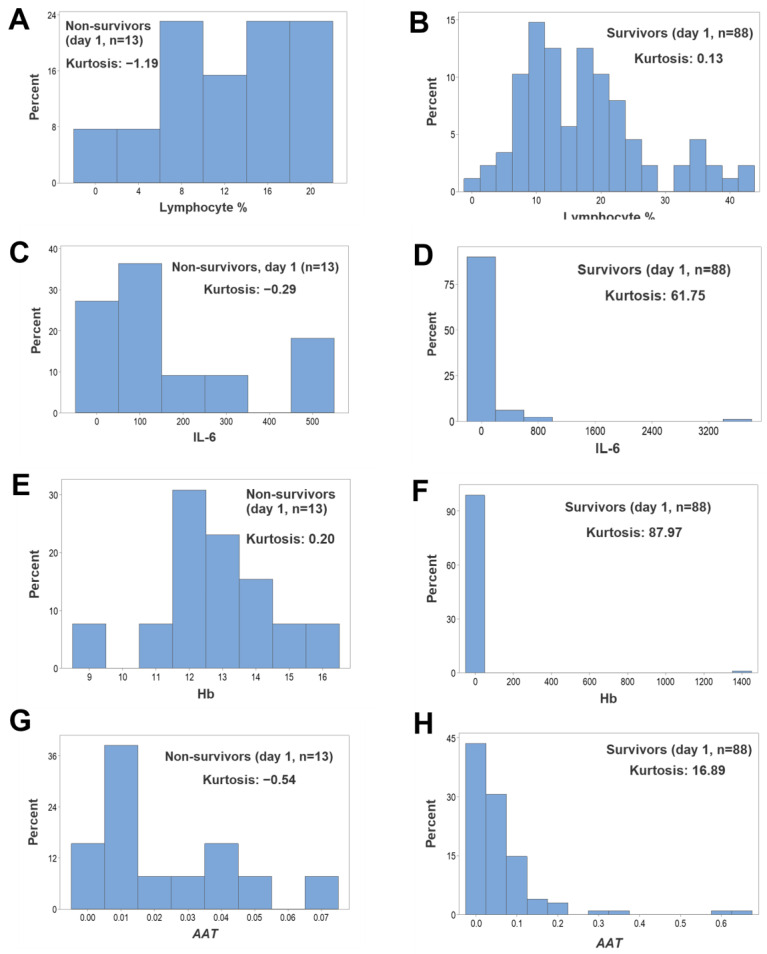
Day 1-specific assessments of kurtosis. Histograms of variables collected on the first hospitalization day did not reveal kurtosis (values equal to or less than 0.13) when, regardless of outcome, non-structured, leukocyte-related data were investigated (**A**,**B**). While non-survivors did not show kurtosis when other variables were investigated (values equal to or less than 0.29, (**C**,**E**,**G**)), the same variables displayed L-shaped patterns when survivors were evaluated (values equal to or higher than 16.89, (**D**,**F**,**H**)).

**Figure 4 biomedicines-12-00871-f004:**
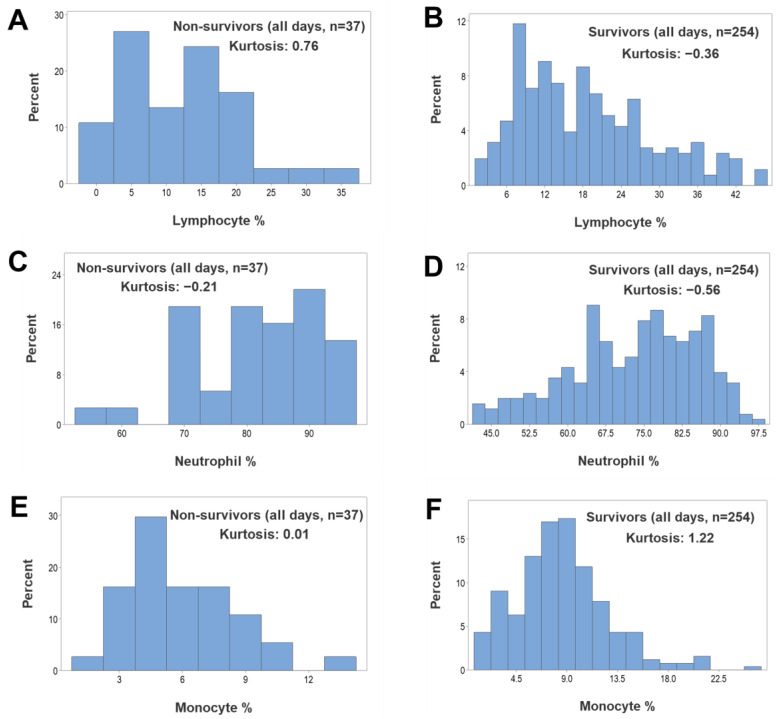
Longitudinal assessments of kurtosis—I. When all longitudinal observations were assessed together (n = 291), no kurtosis was observed (all estimates were equal to or less than 1.22) when, regardless of outcome, the percentages of lymphocytes, monocytes or neutrophils were separately investigated (**A**–**F**).

**Figure 5 biomedicines-12-00871-f005:**
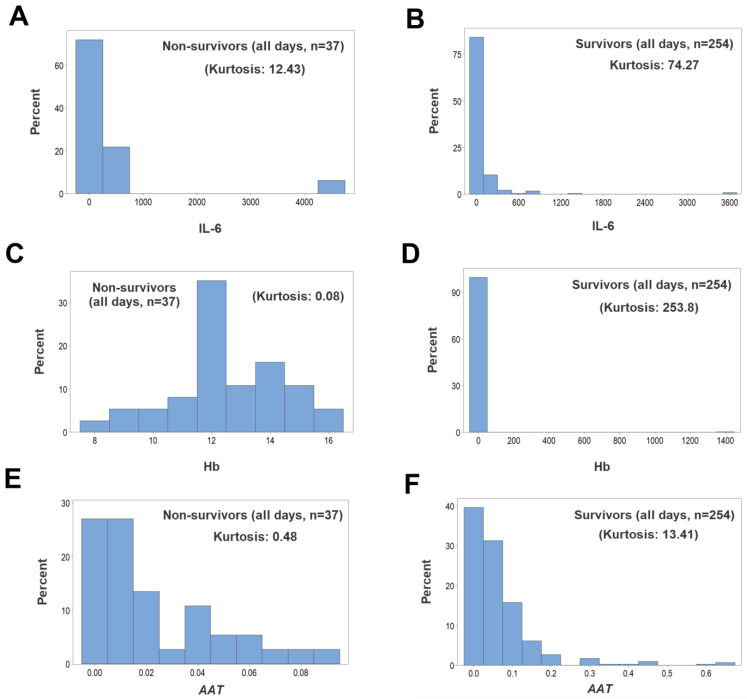
Longitudinal assessments of kurtosis—II. When non-cellular indicators as well as leukocyte-related complex indicators were investigated and all longitudinal observations were considered (n = 291), survivors showed kurtosis estimates ranging from 13.41 to 253.78. In contrast, non-survivors displayed smaller kurtosis estimates (**A**–**F**).

**Figure 6 biomedicines-12-00871-f006:**
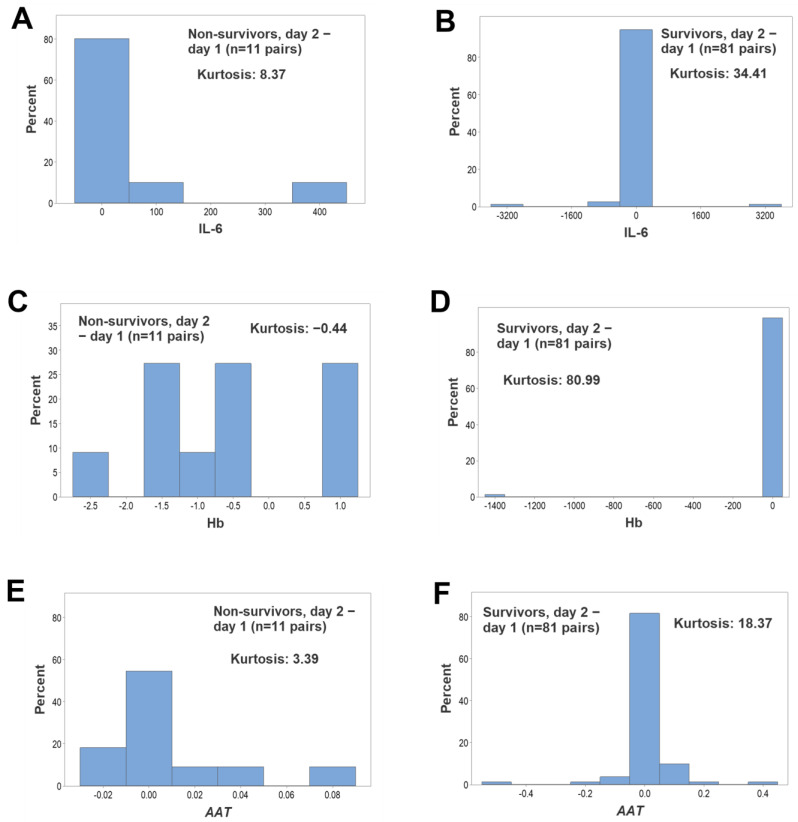
Longitudinal assessments of kurtosis—III (dynamics of paired observations, separate analyses). When the net difference between day 2 and day 1 values of 92 pairs of observations was considered, non-survivors did not show evidence of kurtosis (values ranging between −0.44 and 8.37), while survivors exhibited kurtosis (values between 18.37 and 80.99, (**A**–**F**)).

**Figure 7 biomedicines-12-00871-f007:**
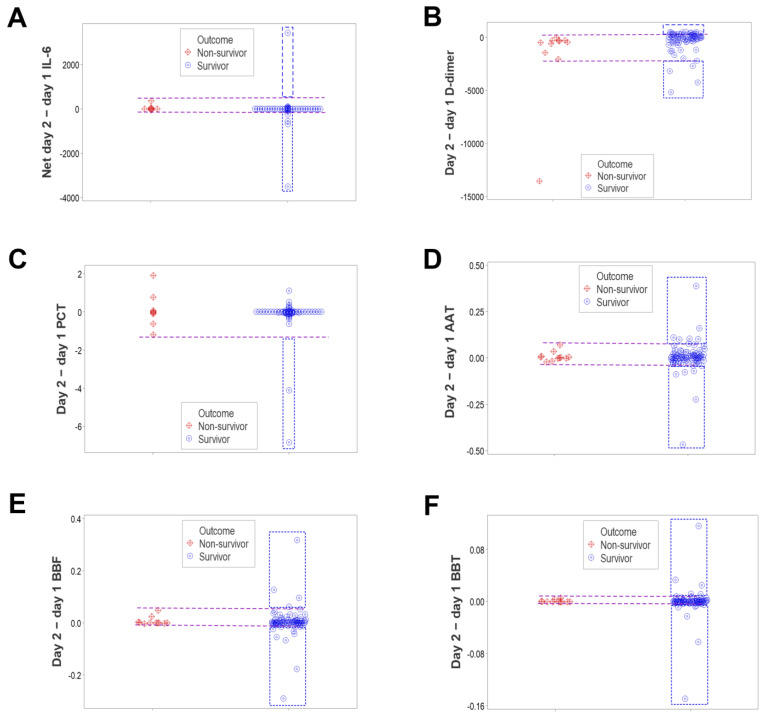
Longitudinal assessments of kurtosis—IV (dynamics of paired observations, integrated analysis). Survivors displayed a larger dispersion of values than non-survivors when the difference between the day 2 and day 1 data values of survivors and non-survivors was simultaneously investigated in a single plot (92 paired observations). Both non-cellular indicators (such as IL-6, D-dimer and PCT, (**A**–**C**)) and complex (structured) indicators derived from leukocyte data (AAT, BBF, BBT, (**D**–**F**)) revealed that non-survivors did not vary much in values over time (the difference of day 2 minus day 1 approached zero). In contrast, all other variables (except PCT) revealed broad oscillations over time when survivors were considered, which expressed both above- and below-zero values. This assessment may be interpreted as a single and perpendicular perspective of the data reported separately for survivors and non-survivors in Figure 6.

**Figure 8 biomedicines-12-00871-f008:**
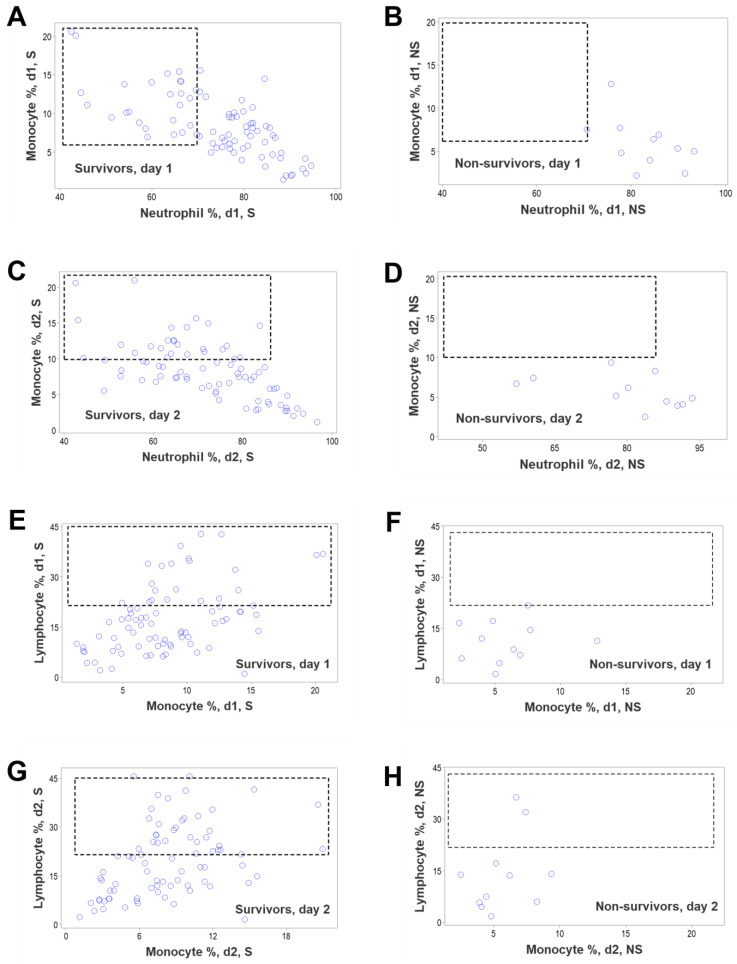
Detection of simple, bi-dimensional relationships associated with survival. While on days 1 and 2, survivors exhibited a broad data interval that facilitated relationships between non-structured variables (neutrophils and monocytes), non-survivors lacked such a data range (rectangles, (**A**–**D**)). A similar absence was observed when lymphocytes and monocytes were assessed (rectangles, (**E**–**H**)).

**Figure 9 biomedicines-12-00871-f009:**
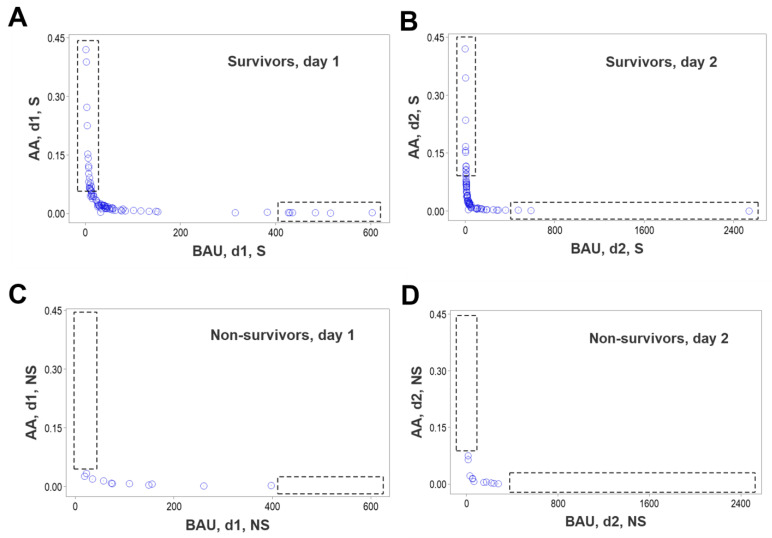
Detection of complex, bi-dimensional relationships associated with survival. When structured, leukocyte-related variables were investigated at days 1 and 2 in survivors and non-survivors, survivors exhibited two data ranges that were lacking when non-survivors were evaluated (**A**–**D**). Because the analysis of non-structured data did not reveal two separate data ranges associated with survival (shown in Figure 8), the analysis of structured (complex) indicators extracted more information from the same data.

**Figure 10 biomedicines-12-00871-f010:**
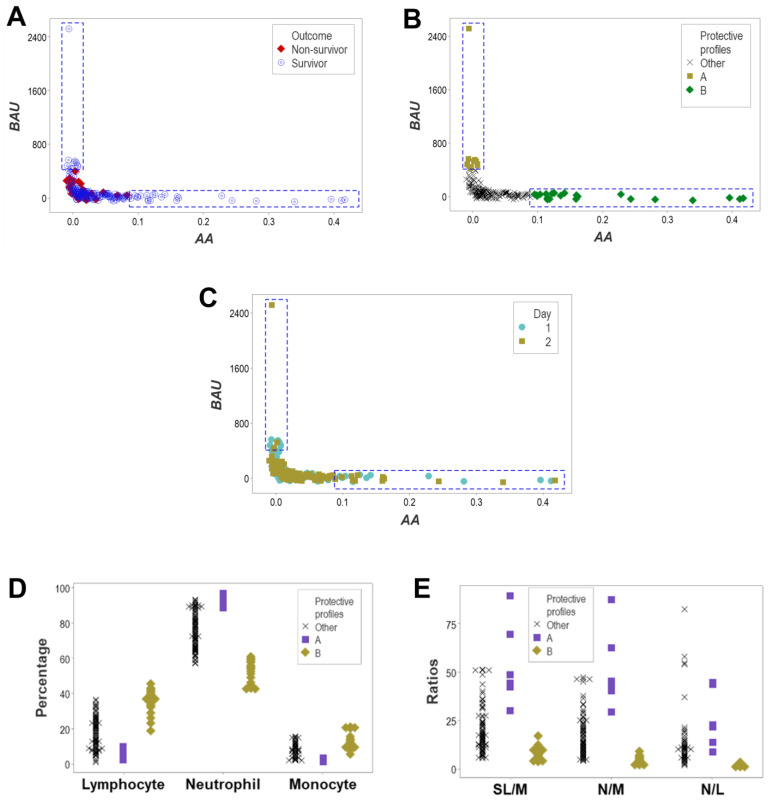
Influence of time on the complexity of bi-dimensional assessments associated with survival. A bi-dimensional, outcome-related, immunological and temporal assessment of structured data is not influenced by time: each of the two data ranges occupied only by survivors included both day 1 and day 2 observations (**A**–**C**). The two subsets of survivor-only observations (here named A and B) were biologically valid and distinguishable: subsets A and B differed from one another both in terms of non-structured data (non-overlapping percentages of lymphocytes, neutrophils and monocytes) and in several ratios, including the small leukocyte (lymphocyte and neutrophil)/monocyte (SL/M), neutrophil/monocyte (N/M) and neutrophil/lymphocyte (N/L) ratios (**D**,**E**).

**Figure 11 biomedicines-12-00871-f011:**
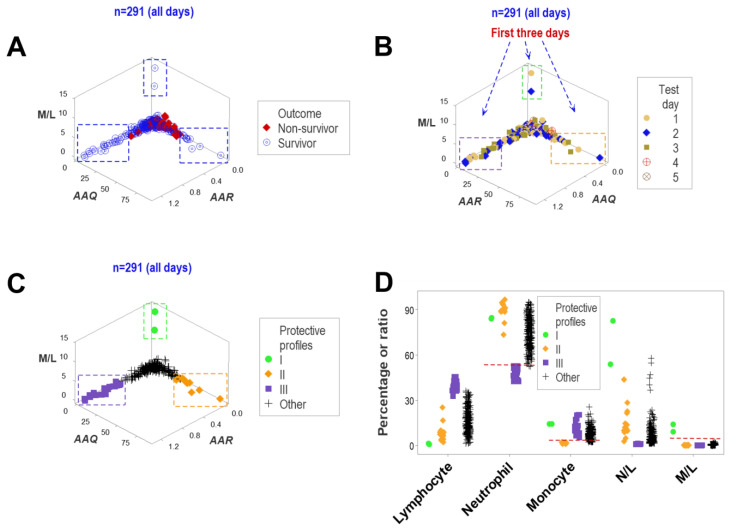
Three-dimensional, long-term, biologically validated detection of structured data ranges associated with survival. More information was retrieved from the same data when three-dimensional relationships were investigated with data collected on all testing days (n = 291). Three non-randomly distributed data subsets—perpendicular to one another—were populated by survivors (**A**). Data oscillations (values far from zero) indicated protective responses if they were observed in the first three hospitalization days (**B**). The combination of spatial and outcome (quantitative and qualitative) data identified three data subsets (named I–III) composed of only survivors (**C**). All three (I–III) data subsets were validated: they differed from one another by non-overlapping intervals of neutrophil or monocyte percentages or monocyte/lymphocyte ratios (**D**). Therefore, immuno-competence was an early response that involved, at least, three complex immunological functions that could involve all cell types.

**Figure 12 biomedicines-12-00871-f012:**
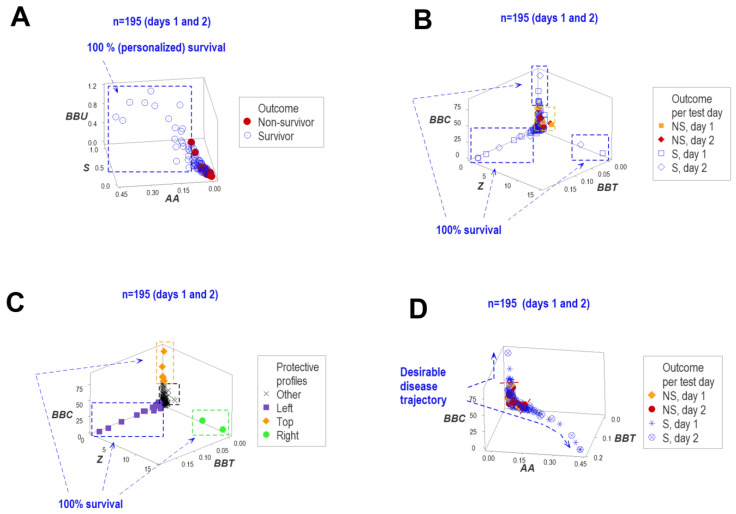
Three-dimensional, two-day long, biologically validated detection of structured data ranges associated with survival. Structured data collected on days 1 and 2 (n = 195) displayed several 3D patterns that facilitated numerous inferences. For example, every data point detected within a data subset perpendicular to the remaining data prognosticated survival (rectangle, (**A**)). Personalized prognoses were supported through redundant analyses: the predictions generated by plot (**A**) were corroborated by the three data subsets distinguished in plot (**B**). The three data subsets identified in (**B**) were biologically valid: each subset associated with survival (‘left’, ‘top’ and ‘right’) differed from one another by two or more variables (**C**). Because this differentiation and validation corroborated the findings reported in Figure 11, it is concluded that the assessment of either day 1 only or days 1 and 2 conveyed similar information. Other structured indicators showed additional potential targets of immunomodulatory therapies and facilitated earlier evaluations of treatments. For example, the temporal data directionality described by a single (one data point-wide) line of observations showed wo desirable trajectories (temporal data movements that followed the arrows, (**D**)). Because this assessment (which was based on data collected over 48 h) corroborated the findings reported both in the analysis of all temporal data and the one that considered only day 1 data, it is concluded than the analysis of complex (structured) leukocyte-related, alone or together with the analysis of other non-cellular hematological variables, may assess immuno-competence, facilitate personalized prognosis and (when temporal data directionality is considered) evaluate therapies earlier.

## Data Availability

Data are contained within the article and Appendix A.

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
