# Peer review of "Decoding Immuno-Competence: A Novel Analysis of Complete Blood Cell Count Data in COVID-19 Outcomes"

_biomedicines, 2024, doi:10.3390/biomedicines12040871_

Round 1
Reviewer 1 Report
Comments and Suggestions for Authors
The article "Decoding immuno-competence: a novel analysis of Complete 2 Blood Cell count data in COVID-19 outcomes" is interesting but difficult to judge because there is little detail in the methods. They mention proprietary software but the citation 40 leads to an abstract in a meeting., acronyms AAT, BBF and BBT are not explained. While I agree on the idea that analysis of data structure can provide better information, it is not clear how could be done in practice.
Author Response
Reviewer 1
The article "Decoding immuno-competence: a novel analysis of Complete 2 Blood Cell count data in COVID-19 outcomes" is interesting but difficult to judge because there is little detail in the methods. They mention proprietary software but the citation 40 leads to an abstract in a meeting, acronyms AAT, BBF and BBT are not explained. While I agree on the idea that analysis of data structure can provide better information, it is not clear how could be done in practice.
Replies to Reviewer I.
Thank you very much for taking the time to review this manuscript. Please find the detailed responses below. To facilitate the review of the revised version, one Word file titled ‘highlighted/tracking changes’ is attached. Unless specified otherwise, the replies described below refer to the new PDF file (‘revised PDF’ file).
A new text is now provided, which explains how the method is conducted (please see new lines 162-173). This description is supported by numerous references, which illustrate how this approach has been applied in several diseases (please see references # 21, 24, 27-29, 39, 40, 44, and 81).
The central concept can be found in Niels Jerne’s 1985 paper, now cited (new reference # 43). He emphasized the language-like properties of immunology, i.e., a set of a few or very few elements (comparable to the 26 letters of the English alphabet) which, once combined and recombined, results not only in a very large number of ‘words’ but an even larger number of ‘sentences’ and/or ‘paragraphs.’ When we refer to Biology, this combinatorial ‘language’ can generate millions of antibodies and/or proteins which can recognize antigens even when the immune system has not previously encountered such antigens and/or perform biological functions.
This means that analyzing only the elemental units (‘letters’, e.g., cell types such as neutrophils) is not informative: following the metaphor on language, no knowledge can be derived from memorizing all the letters of the alphabet. Similarly, useful biomedical knowledge may be missed, unless and until basic leukocyte data are (a) reformatted as information, (b) information is converted into knowledge and, (c) finally, knowledge is applied as concrete (context- and resource-dependent) decisions. This process has been described as the DIKW (data-information-knowledge-‘wisdom’) pyramid (please see reference #29). While a proprietary software can, in principle, facilitate the implementation of all these steps in real-time, any approach that focuses on pattern recognition and investigates data combinations (structures in which the information emerges only after three or more complex variables converge in space) is likely to uncover usable knowledge, as previously described (reference # 24). Please see lines 174-191 of the revised version.
Reviewer 2 Report
Comments and Suggestions for Authors
The article is dedicated to a current medical problem, the SARS-CoV-2 pandemic infection with a high mortality rate. It is also, very promising for early information regarding the immuno-competence of such a patient and his evolution. Using classic investigational data (complete blood count and Il6a) the authors propose a special sophisticated statistical evaluation of structured data, the L-shaped data distribution as positive kurtosis being revealed as a marker for the differentiation of non-survivors from survivors. Visual patterns generated in real-time can inform rapidly on the first day of admission about the real immunocompetence of the patient and predict the evolution, supporting personalization and clinical decisions. This modality of statistical evaluation can serve as a guide to differentiate survivors, who in contrast to non–survivors, are displaying L-shaped data distribution as signs of positive kurtosis, predicting a favorable outcome. They conclude that anticipatory immunocompetence testing may be desirable in SARS-CoV2 infection.
The authors are using a special statistical method, presented as a promising tool for prediction of severe outcome in SARS CoV2 patients.
The authors also underline the limits of the method, accepting the complexity of factors which influence or determine immunocompetence.
Proposals for corrections or changes:
-change „he” in „the” -412/17
-correction of references following general recommendations: after the name of the journal put the year (4,5,6,7,8,9,14,15,16,17,44,48,49,50......)
-it would be useful to introduce conclusions at the end of the article with a short message based on the data of the authors
Author Response
Reviewer 2
Proposals for corrections or changes:
-change „he” in „the” -412/17
-correction of references following general recommendations: after the name of the journal put the year (4,5,6,7,8,9,14,15,16,17,44,48,49,50......)
-it would be useful to introduce conclusions at the end of the article with a short message based on the data of the authors.
Replies to Reviewer 2.
Thank you very much for taking the time to review this manuscript. Please find the detailed responses below. To facilitate the review of the revised version, one Word file titled ‘highlighted/tracking changes’ is attached. Unless specified otherwise, the replies described below refer to the new PDF file (‘revised PDF’ file).
- As indicated, the article ‘the’ has been inserted in former paragraph 412-414 (new line 348 of the revised version).
- Following Biomedicines style, all references have been corrected.
- A new paragraph that starts with a quasi-title (’in conclusion’) highlights the two major conclusions: early and personalized prognostics and immunomodulatory targets that therapies could consider (please see new lines 377-381 of the revised version).
- Other typos were detected and corrected (e.g., new line 348) and the syntax was changed (e.g., lines 76-79). For brevity and/or clarity, text that was not informative or was unclear was deleted or merged (e.g., lines 120-125, 165-166, 202-205, 221-223, 232-234, 255-257, 329-331, 334-347, 374, 378, 382-386, 424 [highlighted version]).
Reviewer 3 Report
Comments and Suggestions for Authors
The present article investigates whether temporal and structured data of the complete blood cell count (CBC) can rapidly estimate immuno-competence.
The authors speak about most published research and they cite just one article- Because most published research on major diseases, such as COVID-19, has been 67 cross-sectional and observational, the risk of bias is not trivial (13).
What were the inclusion and exclusion criteria?
Please specify the limitations of the present study.
In the discussion please include more recent published articles.
The present research does not have a conclusion.
Comments on the Quality of English LanguageModerate
Author Response
Reviewer 3
- The authors speak about most published research and they cite just one article- Because most published research on major diseases, such as COVID-19, has been cross-sectional and observational, the risk of bias is not trivial (13).
- What were the inclusion and exclusion criteria?
- Please specify the limitations of the present study.
- In the discussion please include more recent published articles.
Replies to Reviewer 3
Thank you very much for taking the time to review this manuscript. Please find the detailed responses below. To facilitate the review of the revised version, one Word file titled ‘highlighted/tracking changes’ is attached. Unless specified otherwise, the replies described below refer to the new PDF file (‘revised PDF’ file).
- We appreciate this observation. To substantiate the potential bias of observations studies on COVID-19, we have added three new references (please see new references 14-16).
- Inclusion and exclusion criteria are now explicitly stated. Please see lines 133-139 of the revised version.
- The fundamental limitation revolves about the central concept, on which there is no consensus –immuno-competence. That is why this study analyzed both humoral and cell-mediated responses. Because the data on both types of immune responses supported similar inferences (in both cases survivors exhibited L-shaped [kurtotic] data intervals), the stated limitation was addressed. Please see lines 294-298 of the revised version.
- The current version reports nine references that describe how this approach has been applied in numerous diseases, which involve human and non-human individuals. They are references # 21, 24, 27-29, 39, 40, 44 and 81. Of those, three studies on COVID-19 were published in the last two years (references # 27, 29, 44).
- At the end of the Discussion, this revised version includes a paragraph that starts with a quasi-title (‘in conclusion’). It highlights the two fundamental conclusions (early and personalized prognostics facilitated by a double approach that estimates immune-competence, and the generation of immunological targets or guides that immune-modulatory research could consider). Please see lines 377-381 of the revised version.
- In addition, this version corrected several typos (e.g., line 348) and the syntax was changed (e.g., lines 76-79). For brevity and/or clarity, text that was not informative or was unclear was deleted or merged (e.g., lines 120-125, 165-166, 202-205, 221-223, 232-234, 255-257, 329-331, 334-347, 374, 378, 382-386, 424 [highlighted version]).